# Applications of Nanodiamonds in the Detection and Therapy of Infectious Diseases

**DOI:** 10.3390/ma12101639

**Published:** 2019-05-20

**Authors:** Eva Torres Sangiao, Alina Maria Holban, Mónica Cartelle Gestal

**Affiliations:** 1*Escherichia coli* Group, Foundation of The Health Research Institute (FIDIS), University Hospital Complex (CHUS), of Santiago de Compostela, 15706 Santiago de Compostela, Spain; 2Division of Infection Medicine, Department of Clinical Sciences, Lund University, 221 84 Lund, Sweden; 3Department of Microbiology, Faculty of Biology, University of Bucharest, 030018 Bucuresti, Romania; 4Research Institute of the University of Bucharest (ICUB), 050107 Bucharest, Romania; 5Department of Science and Engineering of Oxide Materials and Nanomaterials, Faculty of Applied Chemistry and Materials Science, University Politehnica of Bucharest, 1–7 Polizu Street, 011061 Bucharest, Romania; 6Department of Infectious Diseases, College of Veterinary Medicine, University of Georgia, Athens, GA 30602, USA

**Keywords:** nanodiamonds, vaccines, microbiology, diagnosis, antibiotic resistance

## Abstract

We are constantly exposed to infectious diseases, and they cause millions of deaths per year. The World Health Organization (WHO) estimates that antibiotic resistance could cause 10 million deaths per year by 2050. Multidrug-resistant bacteria are the cause of infection in at least one in three people suffering from septicemia. While antibiotics are powerful agents against infectious diseases, the alarming increase in antibiotic resistance is of great concern. Alternatives are desperately needed, and nanotechnology provides a great opportunity to develop novel approaches for the treatment of infectious diseases. One of the most important factors in the prognosis of an infection caused by an antibiotic resistant bacteria is an early and rigorous diagnosis, jointly with the use of novel therapeutic systems that can specifically target the pathogen and limit the selection of resistant strains. Nanodiamonds can be used as antimicrobial agents due to some of their properties including size, shape, and biocompatibility, which make them highly suitable for the development of efficient and tailored nanotherapies, including vaccines or drug delivery systems. In this review, we discuss the beneficial findings made in the nanodiamonds field, focusing on diagnosis and treatment of infectious diseases. We also highlight the innovative platform that nanodiamonds confer for vaccine improvement, drug delivery, and shuttle systems, as well as their role in the generation of faster and more sensitive clinical diagnosis.

## 1. Introduction 

Infectious diseases are one of the primary worries of public health systems worldwide. Despite the great expansion in the discovery of new drugs, antibiotic resistance limits treatment options. The misuse of antibiotics, interrupted courses of treatment, and antibiotic usage in agricultural settings, as well as other factors, have had a confounding effect on the increase of resistance. While the majority of the population has a fully functional immune system, the number of immunocompromised patients has substantially increased due to medical advances. In these patients, antibiotic treatment is the only mechanism to fight the infectious threat. An early and accurate diagnosis can significantly increase life expectancy, and nanoparticles (NPs) have become an exceptional alternative to tackle these concerns. Moreover, NPs offer a new approach to treat infectious diseases and, particularly, nanodiamonds are emerging as a great candidate due to their unique qualities [1].

Nanodiamonds (NDs) were discovered in 1963 as a new class of nanoparticles in the carbon family. These nanoparticles, or nanoscale diamonds, are usually smaller than 100 nm and are manufactured by an inexpensive large-scale synthesis based on the detonation of carbon-containing explosives [2]. They were re-discovered in the USSR in 1983 [3]; however, they were not commercially available until 1988 in the USA [4,5]. Currently, carbon-based nanomaterials are being utilized as a drug delivery system because they are well tolerated, and additionally can be used for imaging applications, which makes them exceptionally useful for the care of critical patients [6,7]. NDs are important members of the nanocarbon family; they have a very small size, ranging from 1 to 100 nm [8], allowing for excellent biocompatibility and optical properties [9]. Shortly after their re-discovery, the scientific community began to be interested on their applications in the biomedical field due to their unique characteristics, including versatility and easy manufacturability [10]. The variety of applications for which NDs can be used mainly relies on their chemical production and purification procedures [11,12]. Their use in biomedicine has been significantly increasing in a wide spectrum of applications, including nanoscale magnetic resonance imaging (MRI) cancer therapy [10,13,14,15,16], orthopedic engineering [17], and the synthesis of contact lenses [18]. In addition, NDs show excellent biocompatibility and optical properties useful for microscopy or image diagnosis [19]. 

ND production includes chemical vapor deposition, detonation [4,10], and high-pressure/high-temperature [20] methods (i.e., a bottom-up vs. top-down synthesis approach, respectively) [21]. Different treatment conditions, processing techniques, and production methods generate distinct surface properties resulting in diverse types of NDs that vary in surface chemistry, structure, shape, and size [21,22,23,24], which allows for their classification based on their primary particle or grain size from < 200 nm down to 2 nm [2,21]. 

In the biomedical field, ND detonation is widely used. ND structure can be summarized in a core-shell model, in which the core (the diamond carbon) is inert, while the surface shell is partially graphitic based, allowing for the addition of a variety of functional groups, e.g., carboxyl, hydroxyl [10], or biomolecules such as lysozyme [25], which confers different properties to NDs [26]. Therefore, they can be used as a delivery system for a huge range of drugs, antigens, and antibodies [19]. The remarkable high affinity of NDs with proteins [27,28] enables the generation of a stable and effective conjugate in different buffers, allowing an easy and effective protein load on their surface [27,28,29]. On the other hand, their spectroscopic properties make them ideal for in vivo imaging diagnosis [26], especially for diagnosis of specific targeted cells, increasing the sensitivity of the current therapeutic or imaging diagnosis [4,13,14,15,16,30,31]. In fact, recent advances have highlighted NDs as double-agents combining imaging with drug delivery systems [32] (Figure 1).

In this review, we discuss the properties that make NDs truly unique and extraordinary in comparison to other nanomaterials, focusing on their impact on the medical field, with special attention on infectious disease prevention, diagnosis, and treatment.

## 2. Nanodiamonds as Potential Vaccine Enhancers

Bacteria and viruses have micro-/nano-dimensions [33], and this enhances the hypothetical usage of nanoparticles as a vaccine delivery system or adjuvant, under the premise that they can be processed by the immune system [34,35]. Nanomaterials have revealed intrinsic immunomodulatory properties, being able to act as immune potentiators [7], increasing the immune response. NDs can also be used as co-adjuvants, stimulating the proinflammatory or anti-inflammatory signaling pathways [34,35]. Recently, we face a great variety of medical conditions, including cancer or diabetes mellitus, which are being treated using antibody transfer. In these cases, NDs could be used as a platform to not only deliver the antibodies but also to enhance host immune response [33].

Strong acid-oxidized NDs have a remarkably high affinity for proteins (including antibodies), forming stable conjugates easily and effectively in different conditions via physical absorption [28]. Soluble proteins and native membrane proteins can be easily conjugated onto the surface of NDs after solubilization in detergent micelles, most likely due to the intrinsic hydrophobicity of this carbon-based nanomaterial [29]. Due to their properties, NDs can carry high amounts of proteins; it was proposed that for ~100 nm NDs, a 20–30-µg weight of nanoparticles can carry a 1 µg dried weight of protein [36].

Recent studies reported the preparation of an influenza vaccine based on a mix trimeric H7 (antigenic hemagglutinin motif) antigen with synthetic NDs in an optimized ratio. This nanoconjugate containing the viral protein attached on the surface of synthetic NDs resulted in a virus-like particle vaccine suspension, which was subsequently tested in vitro (hemagglutination assay) and in vivo in a murine model [19]. The obtained vaccine containing the trimeric H7 antigen and synthetic NDs revealed increased efficiency in vitro, resulting in a decrease in the hemagglutination of chicken red blood cells. Moreover, the obtained H7 NDs vaccine produced stronger H7 specific-IgG antibody responses than that with the trimeric H7 [19]. The authors of this study explain the elicitation of a strong and specific immune response of the designed vaccine by an adjuvant effect can be attributed to the NDs. Nonetheless, their results support the idea that NDs provide innovative strategies that can be broadly applied for the development of different vaccines in the future. 

Exploring further the effects of NDs on host immunity, several studies have revealed that IgG antibodies can be adsorbed by modified NDs, which can potentiate their use in several medical settings [37]. NDs possess the ability to bind to Complement component 1q (C1q), a protein of the complement pathway which is involved in many physiological and pathological processes [38], enabling them to modulate host inflammatory signals in an specific manner. 

An area of improvement concerns non-specific biding of the NDs, because after 30 min in the blood system, NDs attached to red blood cell membranes, and they can remain in the circulation without being excreted [39], allowing for their detection in the blood [26,40]. Unfortunately, biodistribution studies in mice revealed that NDs predominantly accumulate in the liver and lungs, although they can also be found in the spleen, kidneys, or even in bone, which could be either beneficial or detrimental for their use [26]. Nevertheless, these deleterious effects can be overcome, and currently there are several research groups working on that.

NDs are highly biocompatible, tunable surface structures that allows for the attachment of other molecules such as drugs or antibodies [41]. Specifically, in the cancer field, ND–antibody (Ab) is presented as a promising approach [42]. Hereby, the integrative properties of NDs make them highly promising for enhancing antibody and drug delivery. 

## 3. Nanodiamonds in Infection Diagnosis

Nanoparticles can be efficiently tailored for the development of useful biomedical tools to be applied in the diagnosis and therapy of diseases, including infections, and this field is rapidly evolving [43]. As mentioned previously, the physical and chemical properties of nanoparticles allow for an accurate, fast, sensitive, and cost-efficient diagnosis [44]. The most important applications and properties of nanodiammonds in infection management are presented in Figure 2. NDs harbor a nitrogen-vacancy enabling them to emit fluorescence when illuminated [9,45,46]; moreover, their magnetic properties can be used as a contrast agent for MRI [46]. Few studies have been carried out in the field of infectious diseases regarding ND diagnosis. One of the first studies, conducted in 2007, proposed a novel method for biolabeling using NDs as detection probes [47]. Using the unique Raman signal of NDs as a detection marker, the researchers were able to visualize biomolecule–bacterial interactions in vivo. Using this technology, the authors were able to detect and localize the position of the interaction between lysozyme and *Escherichia coli* [47].

In 2012, Lin et al. [48] studied the interaction of ciliated eukaryotic unicellular organisms (protist microorganisms), such as *Paramecium caudatum* and *Tetrahymena thermophile*, using different kinds of NDs while testing the relationship between the toxicity and size of NDs. Their results revealed that 5 nm NDs are more toxic than 100 nm ND, probably due to the disordered carbon surface. Furthermore, they assessed the distribution of NDs after injection in *E. coli*, and the results demonstrated that fluorescent nanodiamonds (FNDs) could be used as a bio-label to image any live organism, without any level of toxicity. 

Recently, Soo et al. have developed [49] and validated [50] a strategy for “streamline identification” of *Mycobacterium tuberculosis* complex (MTBC) directly in liquid broth culture media. The authors used a mass spectrometry (MS) approach to analyzed MTBC after culture in BACTEC MGIT 960. By using 5 nm NDs, they reached a limit of detection of 0.09 μg/mL, without albumin interference and avoiding false-positive identifications [49,50]. Hereby, the authors discovered an alternative biomarker of tuberculosis, such as the CFP-10 antigen, and also showed the utility of NDs as efficient probes to be used for the diagnosis of infectious diseases [49,50]. 

An exciting ND-based matrix-assisted laser desorption/ionization coupled with time-of-flight mass spectrometry (ND-MALDI-TOF-MS) approach has also been used by Chang et al. [51] and Zhu et al. [52] to identify a carbapenem-resistant *Acinetobacter baumannii* and human papilomavirus (HPV), respectively. 

## 4. Nanodiamonds in Antipathogenic Systems

Nanotechnology has been used for drug delivery for decades now, and its performance has been highly successful [7,53]. Although, the use of NDs is relatively recent, their small size, high bounding properties, and low cytotoxicity make them highly promising for their use in different areas of microbiology and infectious diseases [39,54,55]. Several published reviews have highlighted the use of NDs for drug delivery, due to its ability to detonate under controlled [56] conditions, which allows for drug release in a controlled manner and in precise locations. However, most of the research has focused on cancer treatment with the goal of developing personalized therapies for cancer patients using NDs in the treatment [6,10,39,56,57,58,59,60,61,62].

NDs also have an intrinsic bactericidal activity [63,64]; Wehling et al. showed that the viability of *E. coli* is nearly 100% compromised after only 15 minutes post-exposure to NDs. This elevated rate of bacteria death was the consequence of a great intake of the NDs by the bacteria, causing deformation of the bacteria cell. Interestingly, the authors demonstrated that there is a direct correlation between the oxygen levels and bacterial death, revealing that the strong bactericidal activity was the consequence of NDs containing partially oxidized surfaces [63]. This particularity of the NDs was further explored by Ong et al. who demonstrated that the bactericidal properties of NDs vary depending on bacteria type (NDs revealed a certain grade of bactericidal activity against *Staphylococcus aureus*), concentration, size, structure, and time of exposure, among others [65,66]. Excitingly, NDs also greatly affect biofilm formation, which is a major problem in healthcare settings. In *S. aureus*, NDs inhibit biofilm formation in a concentration-related manner; however, the results for *E. coli* are contradictory in this regard [66]. 

NDs have been bound to several antimicrobials, including antifungal and antibiotic compounds such as polymyxin B [67], aflatoxin B1 [68], tetracycline, and vancomycin [69]. Remarkably, NDs conjugated with amoxicillin were able to internalize into T24 bladder cells containing uropathogenic *E. coli*, and the results demonstrated that the decrease in bacterial recovery was associated with an increase in ND-amoxicillin treatment dosages [70]. The authors revealed that internalization of these molecules happens in only 2 hours and that internalization is necessary to effectively kill bacteria [70]. The most relevant antimicrobial properties of NDs are summarized in Table 1, and they refer mainly to virulence modulation, biofilm control, growth inhibition, and intracellular pathogen killing.

NDs also have the ability to bind to viruses, such as hepatitis B or C, from blood plasma isolated from infected patients [74], increasing exponentially the applicability of these nanostructures. Thrillingly, in vitro results obtained by Roy et al. [75] demonstrate that NDs can be conjugated with anti-HIV drugs, and due to their low toxicity and small size, these combined particles have the potential ability to cross the blood–brain barrier, increasing the distribution of the drug and reducing the viral load significantly. 

Regardless, more work needs to be done to improve their bio-distribution and toxicity, because for now, NDs can circulate in the blood without being secreted (lower secretion in urine and feces was observed), and unfortunately, they accumulate in the liver (within macrophages) and lung tissues. For this reason, and despite results in animal models being promising [56,76], the potential of NDs remains limited. However, by working on the purification method and its structure, NDs can be extremely improved in their functionality and properties to avoid unspecific reactions by, for example, inclusion in microgels [54,77]. 

Overall, NDs have great potential in the field of infectious diseases. Antibiotic resistance is one of the major threats that current society is confronted with. Although novel antibiotics are being investigated, these are mostly based on modifications of current antibiotics, and the likelihood of resistance is high. NDs offer new opportunities for the treatment of infectious diseases. In the near future, we estimate that more research will focus on nanotechnology, and NDs in particular, for shuttle and drug delivery systems. 

## 5. Conclusions

NDs have great potential for their application in the design of biomedical materials due to their great physico-chemical characteristics and low toxicity. Their intrinsic fluorescence and ability to bind bioactive molecules further promotes their use for different aspects of the biomedical field. NDs possess remarkable mechanical and optical properties and a large surface area, which makes them highly useful in drug delivery and diagnosis approaches. Preliminary data has revealed that NDs have the ability to modulate the host immune response, and this is a key feature to fight against pathogens. Furthermore, the versatile characteristics of NDs make them a great candidate to improve drug delivery, which combined with their fluorescence ability, allows for the monitoring of the drug. The use of NDs in drug delivery has not been tested yet in clinical settings; however, they have yielded promising expectations for new therapies in animal models. Some of the properties that NDs need for future applications include robustness against thermochemical changes in their surroundings and persistence in the “hostile environment” of the host. One of the limitations for medical applications is their size, because sizes smaller than 50 nm can trigger aggregation and subsequently accumulation. Therefore, we need to promote studies of the metabolism and ND-drug clearance from the host. 

Nano-diamonds offer a novel approach to decrease the high levels of co-morbidity and mortality associated with antimicrobial resistance, as well as to decrease the costs of treatment, ultimately leading to the subsequent decrease in antimicrobial resistance. 

## Figures and Tables

**Figure 1 materials-12-01639-f001:**
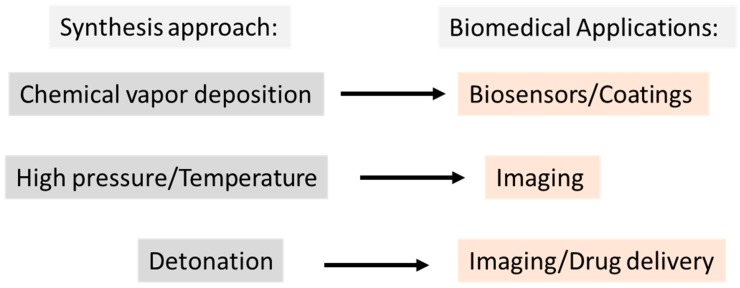
Schematic of the main applications of nanodiamonds based on the synthesis method.

**Figure 2 materials-12-01639-f002:**
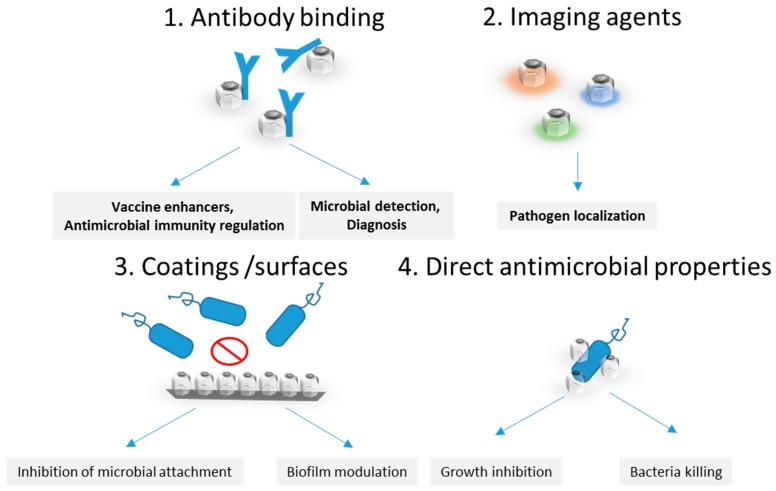
Proposed routes for infection, detection, and antimicrobial therapy to be further investigated.

**Table 1 materials-12-01639-t001:** Main types of nanodiamonds related to their antimicrobial properties and effects.

Type	Antimicrobial Effect	Target Species	Effect was Observed	References
Glycan-modified NDs	Inhibition of type 1 fimbriae-mediated adhesion	*Escherichia coli*	In vitro	[66,71]
ND-NH2, ND-COOH	Biofilm inhibition	*Escherichia coli, Staphylococcus aureus*	In vitro	[72]
menthol modified NDs	Growth inhibition	*Escherichia coli, Staphylococcus aureus*	In vitro	[73]
oxygen-containing surface groups - NDs	Bactericidal properties	*Escherichia coli, Bacillus subtilis*	In vitro	[63]
acid-purified 6 nm NDs	Intracellular pathogen killing	intracellular uropathogenic *Escherichia coli*	In vitro (T24 bladder cells)	[70]

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
