# Peer review of "Applications of Nanodiamonds in the Detection and Therapy of Infectious Diseases"

_materials, 2019, doi:10.3390/ma12101639_

Reviewer 1 Report

The manuscript is now fit for publication. I have one final editorial comment:

Line 256, … one of the major threats the current society is confronted with.

Author Response

Thanks for the help with the grammar and overall for all the constructive comments along this process, the paper has substantially improved and it is because all the constructive comments we have received. We have now changed it to “Confronted with”

Reviewer 2 Report

Although the topic seems interesting and some changes were made, I think the article is still premature, very short and direct, as it lacks critical discussions, figures and convincing outlook. Moreover, in some instances talks about introduction of nanotechnology again in the mid of the review (for instance section 3, Nanoparticles (1 – 100 nm) are.....), suggest rearrangement of the text appropriately and remove some of the redundant expressions as well. No notable figures in the manuscript as NDs have been reported in numerous studies. The authors were predominantly focused on giving the information. However, appropriate figures should be selected and showcased, if possible need to redraw for better insights and convenience to the reader. Citable items should be appropriately placed in the order, but some of them were placed after the references section. follow the journal style and arrange appropriately

Author Response

We would like to thank all the constructive reviews that this reviewer has provided to us. We believe that the manuscript is more focused and clear now that this reviewer had made such an effort to help us to keep focus and concise. 

Although the topic seems interesting and some changes were made, I think the article is still premature, very short and direct, as it lacks critical discussions, figures and convincing outlook. Moreover, in some instances talks about introduction of nanotechnology again in the mid of the review (for instance section 3, Nanoparticles (1 – 100 nm) are.....), suggest rearrangement of the text appropriately and remove some of the redundant expressions as well. 

We have removed that redundancy as well as others in the manuscript. 

No notable figures in the manuscript as NDs have been reported in numerous studies. The authors were predominantly focused on giving the information. However, appropriate figures should be selected and showcased, if possible need to redraw for better insights and convenience to the reader. 

Figure 1 has been replaced for a schematic of the applications of nanodiamonds based on their synthesis. We believe this figure is more descriptive and focused than our previous.

Citable items should be appropriately placed in the order, but some of them were placed after the references section. follow the journal style and arrange appropriately

We would like to especially thank this comment because we realized that we have used different bibliographical styles and databases. We have done the references again to avoid confusion and double numbering.

Reviewer 3 Report

Review for Manuscript materials-499736-peer-review-v1

General Comments: Overall, very interesting topic and thorough discussion of the pluses and minuses of nanodiamond therapy. Almost all of my comments are text edits, listed by line number below.

More Specific Comments:

1)    Line 43 – Change “worrisome” to “worrisomes”

2)    Line 45 – Remove “ruthlessly” and “nowadays”

3)    Line 47 – Change “confound” to “confounding”

4)    Line 48 – Change “Although,” to “While the”

5)    Line 61 – Change “USA” to “the USA”

6)    Line 62 – Change “as drug” to “as a drug”

7)    Line 65 – Remove “specially”

8)    Line 66 – Remove “them”

9)    Line 84 – Change “biomedical” to “the biomedical”

10) Line 89 – Change “as delivery” to “as a delivery”

11) Line 94 – Change “into” to “in”

12) Line 105 – Change “diseases” to “disease”

13) Line 108 – Change “has enhance” to “enhances”

14) Line 112 – Change “immune” to “the immune”

15) Line 129 – Change “in optimized” to “in an optimized”

16) Line 147 – Remove “an”

17) Line 148 – Change “unspecific biding” to “non-specific binding”

18) Line 152-153 – Accumulation in the spleen, kidney and bone is not necessarily detrimental, it could be very important for delivery of therapies to these organs.

19) Line 157 – Change “on” to “in the”

20) Line 167 – Change “[46” to “[46]”

21) Line 172 – Change “detected” to “detect”

22) Line 226 – Change “bounded” to “bound”

23) Line 269 – Change “modulate” to “to modulate the”

24) Line 281 – After “mortality” insert “associated with antimicrobial resistance”

Author Response

We would like to thank the reviewer to take the time and effort in helping us with the editing of the manuscript, we can appreciate the detail effort that this reviewer has put in correcting our precarious En. We appreciate how the effort of this reviewer has significantly improved the quality of the manuscript. Thanks.

1)    Line 43 – Change “worrisome” to “worrisomes”

We have now changed it to worrisomes

2)    Line 45 – Remove “ruthlessly” and “nowadays”

Both words have been removed from line 45

3)    Line 47 – Change “confound” to “confounding”

Cofound has been replaced to cofounding

4)    Line 48 – Change “Although,” to “While the”

We have now While the at the start of the sentence

5)    Line 61 – Change “USA” to “the USA”

We have added the article in line 61

6)    Line 62 – Change “as drug” to “as a drug”

We have added “a”

7)    Line 65 – Remove “specially”

Specially has been removed

8)    Line 66 – Remove “them”

Them has been removed

9)    Line 84 – Change “biomedical” to “the biomedical”

We have added “the”

10) Line 89 – Change “as delivery” to “as a delivery”

We have added “a”

11) Line 94 – Change “into” to “in”

We have changed to in

12) Line 105 – Change “diseases” to “disease”

We have changed the word to singular

13) Line 108 – Change “has enhance” to “enhances”

We have changed the verb

14) Line 112 – Change “immune” to “the immune”

We have added the article

15) Line 129 – Change “in optimized” to “in an optimized”

We have added the article

16) Line 147 – Remove “an”

We have removed the article

17) Line 148 – Change “unspecific biding” to “non-specific binding”

We have changed to non-specific

18) Line 152-153 – Accumulation in the spleen, kidney and bone is not necessarily detrimental, it could be very important for delivery of therapies to these organs.

Changed to “. Unfortunately, biodistribution studies in mice revealed that NDs predominantly accumulate in liver and lung, although they can also be found in spleen, kidney or even bone, which could be either beneficial or detrimental for their use [25].”

19) Line 157 – Change “on” to “in the”

We have changed to in the

20) Line 167 – Change “[46” to “[46]”

We have corrected the mistake

21) Line 172 – Change “detected” to “detect”

We have changed the past tense to present

22) Line 226 – Change “bounded” to “bound”

We have changed the past tense to present

23) Line 269 – Change “modulate” to “to modulate the”

We have changed it to “to modulate the”

24) Line 281 – After “mortality” insert “associated with antimicrobial resistance”

We have now changed it to “subsequent decrease mortality associated with antimicrobial resistance”

This manuscript is a resubmission of an earlier submission. The following is a list of the peer review reports and author responses from that submission.

Round  1

Reviewer 1 Report

The review aims „to  discuss  the  latest  findings  made  on  the  nanomdiamonds  with  a  special  interest  in  prevention,  diagnosis,  and  treatment  of  infectious  diseases.  Highlighting  innovative  the  platform  for  vaccines,  faster  and  more  sensitive  clinical  diagnosis and mainly, shuttle and drug delivery systems.“ As compared to the 2016 review the authors published where they aimed „to discuss the main advances made on the science of nanomaterials for the prevention, diagnosis and treatment of infectious diseases. Highlighting innovative approaches utilized to: (i) increasing the efficiency of vaccines; (ii) obtaining shuttle systems that require lower antibiotic concentrations; (iii) developing coating devices that inhibit microbial colonization and biofilm formation

While individual statements in the review are mostly not false, they become misleading and make no sense when they are put together. The authors have obviously no practical experience with nanodiamonds and hence cannot put together information from other articles properly with the knowledge and experience needed. Authors then often mix together various different aspects (for example Raman scattering, photoluminescence, magnetic resonance sensing); or another example is the use of referencing to origin of nanodiamonds. Authors are not able to distinguish between detonation nanodiamonds and HPHT nanodiamonds – authors do not even mention those even though they refer to their properties and to results obtained using such king of nanodiamonds. Overall, the review seems like mixing of apples and pears – statements and parts of other reviews put together without any comprehensive meaning. I would hesitate to give such review to a student as an introduction to the topic as it would bring more questions than answers.

One example of an obvious inconsistency. In the abstract, authors claim that “The key in factor to overcome infectious diseases is an early and accurate identification of the bacterial strain, and this novel nanodiamonds approach offers this.“ However authors do not mention any bacterial strain detection using nanodiamond in the review and I do not know what kind of „novel nanodiamond approach“ would enable it nor what kind of „strain“ authors mean. There are many more inaccuracies and inconsistencies in the text. On top of that, there are many factual errors that might have been caused by the lack of proof reading/missing words that change the meaning of a statement (an example: p3, l97 – “white cell line”), the word Raman is for some reason written with all capital letters. Number of errors in References section: references are either blended together or divided into two.

I find the text inconsistent and problematic to the extent that goes beyond extensive individual corrections and do not recommend it for publication.

Author Response

We would like to thank this review for highlighting the pitfalls of this manuscript. We acknowledge that we are naïve on some aspects of the design of nanodiamonds particles and we have consulted an expert on this area to overcome this limitation on a manuscript that we consider necessary to compile all the advances that have been accomplished in this area. We have profoundly worked on the language and pass the manuscript to a specialist on the subject. We sincerely hope that the concerns that made this reviewer to advice the rejection of the manuscript has been addressed.

Reviewer 2 Report

The manuscript entitled Nanodiamonds and its applications to Infectious Diseases field summarizes the use of nanodiamonds towards Infectious Diseases. Despite the discussion, I suggest expanding some sections to further improve the article before publishing. Moreover, I suggest language check as numerous grammatical errors have been retained.

Spell error in the title, “ Nanodiamonds.”

The introduction is too short and lacks sufficient background, I suggest introducing the problems associated with the infectious disease in the introduction section, as the authors directly start discussing the usage of nanodiamonds for vaccines and others from section 2. Why only nanodiamonds are appropriate for these applications should be discussed.

I suggest discussing the stand-out points for nanodiamonds that could defend against various innovative nanocontainers, especially inorganic such as Mesoporous silica (Journal of Photochemistry & Photobiology, B: Biology 169 (2017) 124–133, DOI: 10.1021/mp500836w), layered double hydroxides, gold, etc. in the essence of acting against infectious diseases

Most of the instances (section 3 and 4), authors summarized others reports, better give meticulous discussions and your perceptions on the articles maintaining the logic. Moreover, the focus is just based on the prep of nanodiamonds and their utilization but no things like on what strain and what mechanism, etc.

Many sentences are difficult to understand, like the last sentence of the abstract, the first sentence in section 2, better improve writing. Line 88, “likewise nanoparticles, NDs” should we not call ND’s as nanoparticles? I think should change it to ‘other nanoparticles or other inorganic nanoparticles’.

In Line 128, What does nanotechnology-technology refers to?

Needs a table for disclosing the modifications on nanodiamonds for different strains so far or drugs enclosed to deliver.

Redundant italic and underlining (Line 97), need to edit carefully

Line 215, change ‘in’ to ‘as’

Reviewer 3 Report

The manuscript needs extensive language editing before it can be effectively reviewed for its technical scientific content. Shortcomings on pages 1 and 2 (abstract, introduction) comprise: usage of articles (in some instances not present, in other instances redundant), singular and plural not used correctly in the sentence context, mix up of adverb with adjective or vice versa, missing words in sentence (e.g. noun, line 71), wrong form of verb (1st or 2nd participle), missing conjunctions (e.g. of), comma setting.

Specific comments:

Abstract

Line 17, … continues being infectious …

Lines 18-19, invert two sentence parts: Antibiotics are designed to …, however, antimicrobial resistance could cause …

Line 20, estimated 10 million deaths per year for each sepsis and antibiotic resistance. How does "by 2050 relate to this?

Line 22, key factor

Line 24, nano-diamonds are great particles …

Line 27, … such as respective vaccines or drug delivery systems (

Line 29, with special interest in

Lines 30-31, this sentence is incomplete

Introduction

Line 35, nanoparticles

Line 39, EU?

Line 39, the scientific community

Line 41, ease in manufacturability

Line 44, biofilms

Line 45, Currently, carbon-based nanomaterials

Line 48, most innovative nanoparticles?

Line 57, high-pressure high-temperature? A noun is missing here.

Nanodiamonds: defined as sized 1-100nm in line 49 and as size range 1-150 nm in line 60, which is correct?

Ultrananocrystalline particles: defined is size range 2-10 nm in line 61 and as 4-5 nm in line 66, which is correct?

Line 63, addition of

Line 67, promising

Line 78, figure 1, what is "others"? One or two examples in a footnote could be indicated.

Lien 82, systems.

Author Response

Round  2

Reviewer 1 Report

The manuscript "Applications of nanodiamonds in the detection and its applications to therapy of Infectious Diseases" was rewritten greatly and some of the inacuracies were corrected and clarified. 

Even though some progress was made, the manuscript still contains critical amount of inconsistencies and false statements to the extent that when published, I cannot see it would benefit to the readers of Materials journal. To make their point, authors often support their statements by references that do not contain such statement, which results in misleading interpretation and gives false impresion. 

Besides the major conserns, manuscript still contains wast amount of small inacuracies and mistakes in language. 

For example: 

- Line 140 - states that "NDs were discovered in 1963", but later in the text, authors speak about "re-discovery in 1963" - what does it mean?

Line 148 -  "NDs are specially important members of the nanocarbons family, they have a very small size, ranging from 1 to 100 nm [8], allowing them for an excellent biocompatibility and optical properties [9]" - how does a very small size allow for excellent biocompatibility and optical properties? - the statement is somhow duplicate to the one in lines 159-161

Line 197 - "diamonds (< 2nm)" - is clearly wrong

- not all references are cited in the textand many many others (possibly pointed out by other reviewers)

Examples of major mistakes and misleading statements:

Pragraph from 198-201 - is not correct. This might be ok only for detonation diamonds known for their core-shell structure. But as authors state in the previous paragraph, different production and processing methods result in completely different surface properties, where in many, the core-shell model is wrong. Later in the paragraph, authors also without any logic choose to give an example of functional groups "carboxyl, hydroxyl or lysozyme" - why lysozyme? In the cited work, lysozyme (large biomolecule vs functional group) was only non-covalently bound to the surface.

Chapter 2 Nanodiamonds as vaccine enhancers

In the first paragraph, authors write about vaccines and link  them to the interaction of nanoparticles with an immune system. However, the only statement related to nanodiamond: 

270 - "NDs can also be used as co-adjuvants, stimulating the proinflammatory or anti-immunity pathways [33]" - cites a reference that does not contain a word "nanodiamond" at all. The other reference in the paragraph is also not related to nanodiamond. It gives an impresion, that all these things are simply made up by authors, who clearly do not have any support for their claims in the literature.

Next paragraphs of this chapter do refer to some works truly done on nanodiamonds, but they are related to cell line interactions and the use of ND as labels, internalization, the detection methods etc.  I indeed miss any accurate information related to Nanodiamonds as vaccine enhancers.

All this occurs in similar way also in Chapter 3 - Nanodiamonds in accurate infection diagnosis

The first sentence (Line 350): "NDs are being exploit to prevent infectious diseases and this field is rapidly evolving [42]" is again refering to work where nanodiamonds (or diamonds) are not mentioned at all. Again the whole paragrap gives a false impresion that nanodiamonds are already being applied to accurate infection diagnosis. Further, it again only summarizes nanodiamonds as markers. Further in the text, author claim that NDs have exceptional magnetic properties, but in the cited original work, the magnetic properties of "magnetic diamonds" originated only in iron nanoparticles that were conjugated to nanodiamonds.

Errors like this can be found also further in the manuscript and therefore I do not recommend to puhlish this review in Materials.   

Reviewer 3 Report

To a large extent, the manuscript was re-written, with substantially improved its quality. I have a few final comments, mostly editorial ones, see below. Subject to appropriately addressing those, the paper should be published.

Specific comments:

Abstract

Line 65, every third patient that

Chapter 2

Line 265, especially viruses and bacteria.

Lines 345-347, antibody, delivery

Chapter 3

Line 350, exploited

Line 352, allow for

Lien 356, The use of

Line 365, what is the meaning of "the size is reliable"?

Lien 371, infections diseases

Line 453, analyse; what is the D in DND?

Line 459, An exiting ND-based

Chapter 4

Line 506, their hardness

Lien 528, NDs can also influence biofilm formation

Line 579, For this reason, …

Line 586, is confronted with

Lien 589, offer completely new opportunities

Conclusion

Line 649, The use of

Line 654, Nevertheless, there are many challenges